# Potential Chemopreventive Role of Pterostilbene in Its Modulation of the Apoptosis Pathway

**DOI:** 10.3390/ijms24119707

**Published:** 2023-06-03

**Authors:** Omchit Surien, Siti Fathiah Masre, Dayang Fredalina Basri, Ahmad Rohi Ghazali

**Affiliations:** 1Center for Toxicology and Health Risk Studies (CORE), Faculty of Health Sciences, Universiti Kebangsaan Malaysia (UKM), Kuala Lumpur 50300, Malaysia; 2Center for Diagnostic, Therapeutic & Investigative Studies (CODTIS), Faculty of Health Sciences, Universiti Kebangsaan Malaysia (UKM), Kuala Lumpur 50300, Malaysia

**Keywords:** pterostilbene, apoptosis, chemopreventive, cancer, cell death

## Abstract

Cancer incidence keeps increasing every year around the world and is one of the leading causes of death worldwide. Cancer has imposed a major burden on the human population, including the deterioration of physical and mental health as well as economic or financial loss among cancer patients. Conventional cancer treatments including chemotherapy, surgery, and radiotherapy have improved the mortality rate. However, conventional treatments have many challenges; for example, drug resistance, side effects, and cancer recurrence. Chemoprevention is one of the promising interventions to reduce the burden of cancer together with cancer treatments and early detection. Pterostilbene is a natural chemopreventive compound with various pharmacological properties such as anti-oxidant, anti-proliferative, and anti-inflammatory properties. Moreover, pterostilbene, due to its potential chemopreventive effect on inducing apoptosis in eliminating the mutated cells or preventing the progression of premalignant cells to cancerous cells, should be explored as a chemopreventive agent. Hence, in the review, we discuss the role of pterostilbene as a chemopreventive agent against various types of cancer via its modulation of the apoptosis pathway at the molecular levels.

## 1. Introduction

Cancer is the leading cause of death worldwide, responsible for almost one in six deaths [1]. According to the statistics from the Global Cancer Observatory (GLOBOCAN), in 2020 alone, there were 19.3 million new cases of cancer reported. It is estimated that there are 10 million deaths caused by cancer based on the incidence and mortality data from 185 countries worldwide. The most common form of human cancer is female breast cancer, followed by lung cancer, prostate cancer, nonmelanoma of the skin, and colon cancer, which account for 42.6%, almost half, of the total newly diagnosed cancer cases worldwide [2]. The development of cancers can be influenced by many factors, including external factors such as environmental and lifestyle factors, internal stress, and genetic defects [3,4]. However, environmental and lifestyle-related factors have been shown to outweigh the genetic or heritable factors as studies showed that the co-occurrence of cancer in identical twins was low [5,6]. Among all the cancer cases, only 5–10% of cancers were due to inherited genetic factors as the majority of human cancers, about 90–95% of cancer cases, have been linked to exposure to environmental agents and lifestyle-related factors [7]. The lifestyles associated with cancer are cigarette smoking, alcohol consumption, obesity, and exposure to environmental pollutants, infectious agents, and radiation [8,9,10,11]. Other than advances in cancer treatments and early detection strategies, lifestyle modification is also a crucial intervention in reducing the cancer burden. Lifestyle modifications s to reduce the risk of cancer include limiting alcohol consumption and the usage of tobacco products, adopting a healthy diet and undergoing regular physical activity to maintain an ideal body weight [12]. In addition, chemoprevention is one of the promising interventions that has gained increasing attention for reducing the burden of cancer [13]. The carcinogenesis or development of cancer is progressive and involves the transformation of the premalignant to malignant or cancerous state through the multistage process which generally consists of initiation, promotion, and progression. Generally, during the initiation stage when the cells are exposed to a carcinogenic agent, deoxyribonucleic acid (DNA) damage is induced and leads to gene mutations. During the promotion stage, the initiated or damaged cell colony expands through the stimulation of cell proliferation. The progression stage involves genetic and phenotypic changes and may involve further mutations and rapid tumour growth [14]. Hence, multistage carcinogenesis provides a more significant opportunity for a chemopreventive intervention or agent to interrupt the conversion process to the malignant stage [15,16].

## 2. Chemoprevention

The term ‘cancer chemoprevention’ was introduced for the first time by Michael Sporn and his co-researchers in 1975 and they described chemoprevention as using natural compounds to avoid or stop the process of cancer development [17]. Later, the definition of cancer chemoprevention was expanded by Gary Kello to making use of any agents to inhibit, stop, or reverse the process of cancer development that involves the conversion from a non-cancerous state to malignant or invasive cancer [18]. Even though the distinction between the chemopreventive agent and chemotherapy or an anti-cancer drug is not clear, chemoprevention has a different target to combat cancer from that of chemotherapy or anti-cancer drugs because the intervention of a chemotherapy agent is aimed at killing cancer cells after they develop [19]. The chemopreventive agent needs to have no tolerance to toxicity or should have no toxic effects as the chemopreventive agent targets a high-risk population that is healthy. In contrast, the chemotherapy agent is used for cancer patients in which less toxicity or some risk of side effects are justifiable or bearable [20]. Moreover, chemopreventive and chemotherapy agents have been shown to work synergically to increase the effectiveness of cancer therapy and also to minimise the side effects or toxicity of cancer therapy [21]. Chemoprevention approaches can be divided into three categories which include primary, secondary, and tertiary chemoprevention. Primary chemoprevention applies to a healthy population with a high risk of developing cancer. Secondary chemoprevention is specific to those already exposed to carcinogenic agents or who have developed premalignant lesions as secondary prevention is expected to interrupt the progression of premalignant lesions to malignant or cancerous lesions. Lastly, tertiary chemoprevention is for individuals who have developed the disease before and recovered from the treatment as the tertiary chemopreventive approach targets the prevention of reoccurrence and development of new or second primary cancers [22,23]. Based on the mechanism of chemopreventive agents on multistage carcinogenesis, chemopreventive agents can be classified as blocking or suppressing agents. The blocking agents target the initiation of multistage carcinogenesis by preventing or mitigating the effect of DNA damage. Moreover, blocking agents can also contribute to DNA repair mechanisms, and reduce the effect of reactive oxygen and other free radical species via the anti-oxidant property and regulation of phase I and II drug-metabolising enzymes or the reduction in carcinogen uptake. In contrast, suppressing agents act in the later stages of promotion and progression of multistage carcinogenesis to inhibit cell proliferation and stimulation of cell senescence and cell death via apoptosis of initiated or damaged cells to reduce the clonal expansion of mutant cells [22,24]. There are numerous mechanisms or various pharmacological properties that are crucial for a potential candidate to be an effective chemopreventive agent such as anti-proliferative properties, anti-inflammatory properties, an inhibition of angiogenesis and growth factor pathways and cell cycle arrest [25]. In addition, regulating apoptosis pathways by the chemopreventive agent is an important cellular event and a promising novel target of molecular mechanisms for cancer prevention [26].

## 3. Apoptosis

The term “apoptosis” was first introduced by Kerr and his team in 1972 and they described apoptosis as an active and programmed event that involves cell morphological changes, as cells that undergo apoptosis exhibit the formation of a histological structure of an ovoid or spherical shape from the fragmented cytoplasm that consists of condensed nuclei [26]. The word “apoptosis” is derived from ancient Greek, meaning leaves falling off from trees during autumn or flower petals dropping off [27]. In recent years, there has been much progress in the study of apoptosis with more understanding of the mechanisms of apoptosis. Generally, the cell that undergoes apoptosis undergoes three major processes including biochemical changes with the activation of caspases, followed by DNA and protein degradation, and lastly changes in the membrane and the identification by the phagocyte for the phagocytosis of apoptotic cells [28]. Caspases play a crucial role in apoptosis as the main mediators where caspases can either be initiators or executioners (also known as effectors) to induce the process of apoptosis [29,30]. The initiator caspases include caspase-2, -8, -9, and -10, and executioner caspases consist of caspase-3, -6, and -7. Initiator caspases act on the apex part of caspase activation in the cell death pathways of apoptosis; for example, caspase-9 has the ability to activate executioner caspases to cause cell death via apoptosis [31,32]. Caspases are the family of enzymes of cysteine-dependent aspartate proteases as caspases primarily mediate the cleaving of the targeted protein at the carboxy-terminal (C-terminal) sides following aspartate residues, and the activation of caspases is necessary for the induction of apoptosis [33,34,35]. This is because caspases produced in our cells exist in the form of zymogens, inactive enzymes that cannot perform any protease activity prior to activation [36]. Hence, dysregulation of caspase activity in the cell leads to abnormalities in cell death through apoptosis. This condition can initiate many pathological conditions such as cancer, neurodegenerative diseases and autoimmune diseases [37,38,39]. There are two major pathways that initiate or regulate the activation of the caspase to induce apoptosis which are the extrinsic pathway via the death receptor and the intrinsic pathway via the mitochondria [40]. Another less common initiation pathway of caspase activation in apoptosis is the intrinsic endoplasmic reticulum pathway [41].

The extrinsic pathway of apoptotic cell death is initiated at the extracellular plasma membrane of the cell when there is a binding of an extracellular ligand such as the Fas ligand (FasL) to the death receptor (Fas receptor) located on the cell surface transmembrane to induce protein structural changes in the receptor via oligomerisation [42,43,44]. This causes the assembly of proteins such as the Fas-associated death domain-containing protein (FADD), an adaptor protein that binds to the receptor located in the intracellular domain and further promotes the activation of initiator caspases such as caspase-8 or -10 via caspase dimerisation. All these events form the multiprotein-signalling complexes known as death-induced-signalling complexes (DISCs) [45,46]. After the activation of initiator caspases caspase-8 and -10 via the DISCs, the activated initiator caspases are capable of activating the executioner caspases such as caspase-3 through a process called cleavage. The pathway is effective enough to cause cell death in some types of cells that are classified as type I apoptotic cells such as thymocytes. However, in other types of cells such as type II apoptotic cells, caspase-8 needs to be involved in an intrinsic pathway via the mitochondria-dependent pathway to cause apoptotic cell death such as that of liver cells [47,48,49].

The intrinsic apoptosis pathway also known as the mitochondrial pathway causes cell death via the caspase-dependent pathway that is triggered by the intrinsic stimulus. There are various stimuli or inducers of “cell stress” that can induce cell death via the intrinsic apoptotic pathway such as chemotherapeutic drugs, exposure to ultraviolet (UV) and gamma (ϒ) radiation, hypoxia, viral infections, depletion of some growth factors, hormones, or cytokines [50,51]. The intrinsic apoptosis pathway is associated with mitochondria as the process of mitochondrial permeability transition (MPT) and the release of apoptotic factors from the mitochondria are needed in this pathway. The intrinsic apoptosis pathway is regulated by the pro- and anti-apoptotic proteins of the Bcl-2 family [52]. The pro-apoptotic proteins of the Bcl-2 family can be classified into two groups based on the structure, the first group comprising multidomain proteins with three Bcl-2 homology (BH) domains (BH1, BH2, and BH3) and the second group comprising “BH3-only” family proteins that consist only of the BH3 interaction domain. The Bcl-2-associated X-protein (Bax), Bcl-2-homologous antagonist/killer (Bak1), and Bcl-2-related ovarian killer (Bok) are grouped with pro-apoptotic multidomain Bcl-2 proteins, and examples of “BH3-only” proteins are the Bcl-2-interacting mediator of cell death (Bim), Bcl-2-associated agonist of cell death (Bad), BH3-interacting domain death agonist (Bid), Bcl-2-interacting killer (Bik), Bcl-2-modifying factor (Bmf), p53-upregulated modulator of apoptosis (Puma), phorbol-12-myristate-13-acetate-induced protein 1 (Noxa), and activator of apoptosis harakiri (Hrk) [53]. The initiation of the intrinsic apoptosis pathway activates BH3-only proteins to bind to the pro-survival proteins in response to the stress stimulus to release BAX and BAK or either one [54,55]. However, BH3-only proteins can also activate multidomain pro-apoptotic Bcl-2 proteins by directly binding to them [56,57]. After the activation of BAX and BAK, they form oligomers to cause the formation of pores on the mitochondrial outer membrane and this leads to the release of pro-apoptotic proteins such as cytochrome c, the second mitochondria-derived activator of caspase (SMAC)/direct inhibitor of the apoptosis-binding protein with a low pI (DIABLO), and endonuclease G from the mitochondrial intermembrane space into the cytoplasm [58,59]. Next, it allows the binding of cytochrome c to the apoptotic protease-activating factor-1 (Apaf-1) and oligomerization happens to form an apoptosome where the recruitment and activation of pro-caspase-9 occurs [60]. The activated caspase-9 can now perform proteolytic activity to cleave downstream effector caspases such as caspase-3 and -7 to activate them [61]. The effector caspases induce various processes related to the cell’s morphological changes during apoptosis, including cell shrinkage, the breaking up of DNA into fragments, chromatin condensation, and changes to the membrane [62,63]. The SMAC plays a crucial role in inducing apoptosis by opposing the inhibitory effect on caspases as the SMAC can suppress various inhibitor apoptosis proteins (IAPs) such as the X-linked inhibitor of apoptosis proteins (XIAP), and survivin, a cellular inhibitor of apoptosis protein 1 (c-IAP1) and 2 (c-IAP2) [64,65]. Figure 1 shows a schematic diagram of initiator and executioner caspase activation via the extrinsic or intrinsic apoptosis pathways.

### Role of Apoptosis in Chemoprevention

Apoptosis is involved in both physiological and pathological processes as apoptosis is needed for normal growth and development and also the abnormality in apoptosis can lead to disease development or affect the treatment outcomes of certain diseases [66]. For example, apoptosis is one of the critical cellular events in cancer, as the ability of cells to avoid or escape from apoptosis is identified as one of the hallmarks of cancer [67]. Uncontrolled cell proliferation and suppression of apoptosis are common conditions in human cancers [68]. In response to the stimulus that can initiate cancer such as DNA damage, cells undergo cell cycle arrest to let the DNA repair process be performed. However, if the damage is massive and is not within the cell’s capacity to be repaired, this can lead to cell death via apoptosis to remove the damaged cells. DNA damage-induced apoptosis is one of the mechanisms of preventing the development of cancer and the failure of this process can contribute to carcinogenesis [69,70,71]. Apoptosis is a critical cellular event in chemoprevention as apoptosis can eliminate genetically damaged or abnormal cells and prevent them from proliferating to serve as a defensive mechanism against cancer development [72]. The elimination of mutated or premalignant cells via apoptosis is preferable in preventing cancer development as apoptosis does not elicit an inflammation response. This is because cell death via apoptosis does not involve the leakage of cell contents into extracellular environments or adjacent tissues. The fast phagocytosis of apoptosis cells acts to avoid secondary necrosis and the production of no inflammatory cytokines by the phagocytic cells [73]. The induction of apoptosis in the premalignant lesion by the chemopreventive agent has been shown to inhibit the progression of premalignant lesions of cancer in azoxymethane-induced colon carcinogenesis in a rodent model [74]. The inhibition of apoptosis has been proven to be one of the mechanisms in cancer development as avoiding or escaping from apoptosis contributes to the transformation of damaged cells into malignant cells and promotes tumour growth [67]. In normal tissue in the homeostatic state, cell populations are maintained by a balance between cell proliferation, differentiation and death. An imbalance of these cellular events can disrupt the homeostatic state and lead to the abnormal growth of mutated cells which eventually can be followed by carcinogenesis [75,76]. Cancer cells have the ability to avoid cell death via apoptosis by disrupting apoptotic pathways, for example via the upregulation or downregulation of the expression of pro- or anti-apoptotic genes and proteins [77]. Animal studies using mouse genetic models have shown that genetic defects in apoptotic proteins such as caspases and BH3-only proteins can block the stimulus of apoptosis and lead to rapid tumour development and growth [78,79].

Hence, targeting the apoptosis pathways in cancer chemoprevention is a hopeful and promising intervention to prevent cancer as many potential chemopreventive agents have been revealed to promote apoptosis [25]. Previously, numerous potential chemopreventive agents were studied extensively and natural products were shown to target the apoptosis pathway in preventing cancer development [80]. For example, a diet containing curcumin, a natural compound from *Curcuma longa* L., showed a chemopreventive effect against colon carcinogenesis in a rat model through a reduction of the incidence and multiplicity of adenocarcinomas. Moreover, curcumin increased the apoptotic index in colon tumours [81]. Another example of a natural compound that has been reported to exhibit a chemopreventive effect via the modulation of apoptosis is honokiol from the Magnolia tree. The oral administration of honokiol significantly reduced the formation of squamous cell carcinoma (SCC) and increased the percentage of the area of normal histology in the lung SCC mouse model. The oral administration of honokiol induced cell death via apoptosis in lung tumours through the release of cytochrome c from the mitochondria and increasing the expression of cleaved or activated caspase-3 [82]. In addition, many other natural compounds including epigallocatechin gallate (EGCG), genistein, luteolin, lycopene, ellagic acid, and lupeol have been extensively studied for their chemopreventive and anti-cancer effects against various types of cancer. The molecular targets of most of these compounds for their chemopreventive and anti-cancer effects are their anti-proliferation, induction of apoptosis, cell cycle arrest and anti-inflammation [83]. Pterostilbene is another natural compound that is widely studied; it has drawn increasing attention as a potential chemopreventive agent and has the ability to regulate apoptosis in various types of cancer [84,85]. However, in this review, we specifically focus on the potential roles of pterostilbene as a chemopreventive and anti-cancer agent, focusing on the regulation of apoptosis and other pathways.

## 4. Pterostilbene

Pterostilbene or 3,5-dimethoxy-40-hydroxystilbene is a natural compound, that belongs to the stilbene derivative under the class of polyphenolic compounds with a phenolic unit and no nitrogen-based functional groups in their structure group, that is produced via the phenylpropanoid and/or the polyketide pathway by plants [86]. Stilbenes can be synthesised by various plant species including plants consumed by humans or used as traditional medicinal herbs [87]. For example, the various stilbene compounds including polydatin, resveratrol, 2,3,5,4′-tetrahydroxystilbene-2-O-β-D-glucoside and pterostilbene have been detected in the traditional Chinese medicine Radix Polygoni Multiflori [88,89]. Pterostilbene is produced in plants as a secondary metabolite and it was isolated for the first time from the heartwood of *Pterocarpus santalinus* also commonly known as red sandalwood [90]. Pterostilbene is found abundantly in other plants that belong to the Pterocarpus genus such as in the tree wood of *Pterocarpus marsupium* that has been used traditionally as a herbal medicine in India to treat diabetes [91]. Apart from that, pterostilbene can be found in natural sources, including blueberries, peanuts, and fruits of *Vitis* species including *Vitis vinifera* and *Vitis amurensis* [92,93,94]. Moreover, the buds of *Vitis amurensis* have been reported to contain high amounts of polyphenolic compounds including phenylpropanoids, anthocyanin, and flavonoids [95]. Pterostilbene has been demonstrated to exhibit good pharmacokinetic properties compared to other stilbene compounds such as resveratrol. Both pterostilbene and resveratrol are the well-known and widely studied stilbenes that have an almost similar chemical structure to that of other stilbenes, consisting of a C6-C2-C6 and two phenyl rings bonded by an ethene double bond [96]. The biosynthesis of pterostilbene from the resveratrol can occur both in vitro and in planta in grapevine (*Vitis vinifera*) leaves that are catalysed by the resveratrol *O*-methyltransferase under a stress stimulus such as exposure to UV light and fungal infection (*Plasmora viticola)* [97]. The biosynthesis of pterostilbene can occur via the shikimate pathway that involves the amino acids phenylalanine or tyrosine. In the shikimate pathway, these two amino acids are converted to p-coumarate and then to p-coumaroyl-CoA, a stilbene precursor. The stilbene synthase catalyses the conversion of p-coumaroyl-CoA into resveratrol. Lastly, *O*-methyltransferase converts resveratrol into pterostilbene via the methylation of two hydroxyl groups of resveratrol [98]. However, the difference in the presence of methoxy and hydroxyl groups attached to their phenyl rings makes pterostilbene possess better pharmacokinetic properties than resveratrol does [99]. The presence of two methoxy groups on the A phenyl ring and one hydroxyl group on another benzene ring was attributed to the higher lipophilicity, which led to an increase in cellular uptake compared to that of resveratrol without the presence of methoxy groups, but with three hydroxyl groups on the phenyl ring [100]. Figure 2 shows the differences in the chemical structures of resveratrol and pterostilbene. Furthermore, pterostilbene has been proven experimentally to display more desirable metabolic stability and greater oral bioavailability as pterostilbene absorption in the body happens more quickly and can be distributed throughout the body [101,102]. Recent data from human and animal model toxicity studies [103,104] revealed that pterostilbene is relatively safe to consume and does not cause severe toxicity or adverse effects. The daily dietary intake of pterostilbene in Swiss mice in an amount of up to 3000 mg/kg of body weight/day caused an increase in the red blood cell and haematocrit count. However, the serum biochemical tests on proteins, electrolytes, and liver and kidney enzymes were not significantly increased. In addition, histopathological examinations of the liver, spleen, kidneys, heart, brain, lungs, and pancreas revealed that the oral intake of pterostilbene did not cause toxicity-related changes or alterations [104]. In addition, a clinical study revealed that pterostilbene is safe to be consumed in amounts of up to 250 mg/day in adults as the biochemical adverse drug reactions (ADRs) reported on the liver, kidney, and glucose markers were not significant when compared to those reported for the placebo group [103].

Pterostilbene has become increasingly popular in medical research and it has been studied extensively for its human health benefits. The medicinal benefits of pterostilbene including its protective effects against the mycotoxin Fumonisin B1 (FB1) induce cytotoxicity in porcine alveolar macrophages (3D4/21) by inhibiting the activation of the JAK/STAT-signalling pathway. Moreover, pterostilbene has been reported to alleviate FB1-induced inflammation by downregulating the gene expression of pro-inflammatory mediators of interleukin-6 (IL-6), interleukin-1β (IL-1β), tumour necrosis factor-alpha (TNF-α), and interferon-gamma (IFN-γ), and to reduce oxidative stress as pterostilbene treatment reduced the reactive oxygen species (ROS) level [105]. Pterostilbene has also been reported to exhibit a cardioprotective effect against endoplasmic reticulum (ER)-induced heart injury via the antioxidant activity and activation of sirtuin-1 (SRT1) to prevent cardiomyocyte cell death [106]. Pterostilbene is capable of regulating the proliferation and differentiation of human periodontal ligament stem cells (hPDLSCs). Moreover, pterostilbene exerts an anti-inflammation effect to protect against TNF-α induced damage on hPDLSCs by reducing the levels of pro-inflammatory mediators of IL-6 and IL-1β and preventing the activation of NF-κβ (p65) [107].

### 4.1. Pterostilbene as a Chemoprevention Agent via Its Modulation on the Apoptosis Pathway

Recently, pterostilbene has been studied extensively and has been proven to exhibit various pharmacological properties that make it a promising candidate to be developed as a chemopreventive agent, such as the properties of anti-inflammation, antioxidation, anti-proliferation, and detoxifying enzyme enhancement [108,109,110,111]. Numerous studies were carried out and revealed the potential of pterostilbene as an anti-cancer agent against the various types of cancer cell lines via in vitro and/or in vivo models. However, in this review, we only summarised the chemopreventive effects of pterostilbene via the modulation of apoptosis in preclinical studies using animal models. Pterostilbene has been shown to inhibit azoxymethane (AOM)-induced premalignant lesions of aberrant crypt foci (ACF) and the development of colorectal cancer in the rat model. Diets containing pterostilbene at 50 and 200 parts per million (ppm) doses significantly reduced the number of ACF in the colon for the premalignant mouse model in 6 weeks of treatment. The molecular investigation showed that a diet containing pterostilbene has an apoptosis-inducing property in the ACF. The ACF of mice from pterostilbene-treated groups showed an increase in the expression of pro-apoptotic proteins including Fas, Fas L, Bax, and Bid, and of apoptotic caspases such as cleaved caspase-9, -8, and 3 when compared to the control groups. In addition, a diet containing pterostilbene at the same dose of 50 and 200 ppm significantly reduced the number of adenomas in the AOM-induced colorectal cancer mouse model for 23 weeks [112]. Pterostilbene administered at the low doses of 50 mg/kg and at the high dose of 250 mg/kg via intraperitoneal injection significantly reduced the tumour multiplicity, volume, and burden in a urethane-induced lung adenoma mouse model. The tumour multiplicity was reduced by 26% for the low dose and by 49% for the high dose of pterostilbene when compared to that of the cancer control group. The tumour volume was also significantly reduced by 25% for the low dose and by 34% for the high dose of pterostilbene in the treated groups. The tumour burden for each mouse was also reduced after the administration of both low and high doses of pterostilbene by 45% and 63%, respectively. Pterostilbene has been shown to activate the executioner caspase caspase-3 as low and high doses significantly increased the protein expression of cleaved caspase-3 in lung tissues compared to that of the control group [113].

A diet supplemented with pterostilbene at 10 mg/kg exhibited chemopreventive effects against prostate cancer development in a transgenic mouse model with phosphatase and tensin homolog (Pten) loss. Mice with a diet containing pterostilbene showed a smaller prostate gland size than those without pterostilbene in their diet did. Moreover, all the mice in the control group without a pterostilbene diet developed high-grade prostatic intraepithelial neoplasia (PIN) or a premalignant lesion of prostate cancer with around 20% of prostate glands involved. However, the pterostilbene diet caused a 50% reduction in the formation of PIN in the prostate group compared to the group without the pterostilbene diet as only around 10% of prostate glands developed PIN. The pterostilbene diet was shown to induce apoptosis as the protein expression of pro-apoptotic protein p27 and cleaved caspase-3 in prostate glands significantly increased in the pterostilbene diet group compared to that in the control group [114]. In the diethyl nitrosamine (DEN)- and carbon tetrachloride (CCl_4_)-induced hepatocellular carcinoma (HCC) mouse model, the administration of pterostilbene at the dose of 100 and 200 mg/kg of pterostilbene via intraperitoneal injection inhibited tumour growth as the number and the maximum size of tumours was significantly reduced in the pterostilbene-treated groups. Pterostilbene also protects the liver from injury by reducing liver markers including aspartate aminotransferase (AST), alanine transaminase (ALT), lactate dehydrogenase (LDH), and alkaline phosphatase (ALP). Pterostilbene has also been shown to increase apoptotic activity by significantly increasing the percentage of apoptosis cells. Further investigation revealed that pterostilbene also increased the protein expression of tumour suppressor protein p53 in the HCC mouse model [115]. The upregulation of p53 by pterostilbene can be associated with its apoptosis-inducing property in the HCC mouse model. This is because the induction of apoptosis by p53 via the direct transcriptional activation of pro-apoptotic BH3-only proteins such as PUMA and NOXA is suggested to be one of the important mechanisms in destroying cancer cells of anti-cancer drugs that target the activation of p53 [116]. In the SCC mouse model induced by N-nitroso-tris-chloroethylurea (NTCU), intraperitoneally administrated pterostilbene at doses of 10 and 50 mg/kg of body weight was reported to delay lung SCC carcinogenesis based on a histopathological analysis. Immunohistochemical staining of lung tissues showed that the expression of cleaved caspase-3 was significantly increased in pterostilbene-treated groups [117]. Table 1 shows a summary of pterostilbene as a chemopreventive agent against different types of cancer in an animal model via its modulation of apoptosis.

### 4.2. Pterostilbene as an Anti-Cancer Agent Acting via the Modulation of the Apoptosis Pathway

On top of having a chemopreventive effect, pterostilbene has also been reported to exhibit an anti-cancer effect via its modulation of apoptosis in many types of cancer. An in vitro study showed that pterostilbene exhibited an anti-cancer effect on the human lung squamous cell carcinoma cell line, H520, and further investigation revealed that pterostilbene could cause apoptosis and cell cycle arrest. The apoptotic caspase activity of caspase-3, -8 and -9 were decreased in a dose-dependent manner under pterostilbene treatment. Pterostilbene treatment also increased the protein expression of pro-apoptotic proteins of cytochrome-c and BAX. However, the anti-apoptotic protein Bcl-2 was reduced in pterostilbene-treated H520 [118]. A treatment with pterostilbene in oestrogen receptor-positive T-47D and triple-negative MDA-MB-231 breast cancer cell lines showed the apoptosis-inducing effect. Pterostilbene treatment on the breast cancer cell line caused an increase in caspase-3 activity and an overexpression of the Bax protein [119]. The apoptosis-inducing property of pterostilbene was also reported in the cervical cancer cell line of HeLa as pterostilbene treatment on cells increased the percentage of apoptotic cells. Pterostilbene treatment on HeLa cells significantly increased the protein expression of activated caspase-9 and -3 and the anti-apoptotic proteins Bcl-2 and Bcl-xl were significantly reduced by pterostilbene treatment [120]. An in vitro study on glioma cells revealed that pterostilbene induced apoptosis via the activation of protein kinase (ERK) 1/2 and phosphorylated c-Jun n-terminal kinase (JNK) pathways. Pterostilbene treatment on glioma cell lines (T98G and LN18) increases the protein expression of pro-apoptotic proteins Bax, cleaved PARP-1 and cleaved caspase-3 and -9. Pterostilbene also downregulates the protein expression of anti-apoptotic proteins of Bcl-2 and survivin [121]. Figure 3 shows the summary of pterostilbene and the anti-apoptotic and pro-apoptotic mediators that lead to its anti-cancer and chemopreventive effects.

### 4.3. Other Molecular Mechanisms of Pterostilbene as an Anti-Cancer and Chemopreventive Agent

In addition to the pterostilbene’s modulation of the apoptosis pathway, pterostilbene has been shown to target other pathways in its anti-cancer and chemopreventive effects. Cell proliferation and cell death are mainly associated with carcinogenesis as the imbalance between these two cellular events is regarded one of the cancer hallmarks [68]. The anti-cancer and chemopreventive effects of pterostilbene are induced via the regulation of cell proliferation, which is one of the most widely studied pathways. Pterostilbene showed anti-proliferation activity against melanoma cancer cell lines as pterostilbene induced cell cycle arrest in the S phase, the upregulation of the gene expression of CCND1 encoded for the cell cycle protein cyclin D1, and the upregulation of the cell cycle inhibitor gene expression of CDKN1A encoded for the cyclin-dependent kinase (CDK) inhibitor of p21 [122]. An in vivo study on colon cancer cell lines Caco-2 and HT-29 showed that pterostilbene exhibits anti-proliferation activity as pterostilbene induced cell cycle arrest in the transition from the G2 to M phases [123]. Another study on HT-29 colon cancer cells also reported that pterostilbene decreased the proliferation index of Brdu and the induction of cell cycle arrest from G1 to G0. In addition, the gene expression of CCND1 was reduced by pterostilbene and the gene expression and protein level of p21 were increased after pterostilbene treatment. Pterostilbene was also reported to induce apoptosis as it increased caspase-3 activity, DNA fragmentation, and the gene expression of BAX. At the higher dose, pterostilbene could regulate autophagy in HT-29 colon cancer cells as pterostilbene increased the gene expression of ULK1 that is responsible for initiating autophagy and increasing gene expression of AMBRA1 and LC3A which are encoded for proteins that are involved in autophagosome formation [124].

However, in pancreatic ductal adenocarcinoma (PDAC) cell lines, pterostilbene in combination with chloroquine was reported to prevent autophagy and led to the inhibition of PDAC growth. Other than the inhibition of autophagy, pterostilbene has been shown to suppress the pathways of RAGE/STAT3 and protein kinase B (AKT)/mammalian target of rapamycin (mTOR) and lead to the induction of apoptosis in PDAC [125]. However, pterostilbene treatment alone without chloroquine showed an autophagy induction effect by the upregulation of the expression of the LC3-II and Beclin1 proteins. Autophagy has both tumour-suppressing and -promoting roles depending on a few factors including the external stimuli, type of cell, cancer progression or stage, and components or ecosystems of the surrounding tumour. For example, the effect of autophagy in tumour suppression includes the protection of DNA from damage and the inhibition of ROS production during carcinogenesis. In contrast, autophagy also creates tumour-promoting effects by supporting cancer cell survival via the inhibition of apoptosis and the stimulation of cell proliferation and resistance toward chemotherapeutic agents [126]. In gastric cancer, pterostilbene has been shown to enhance the effectiveness of sunitinib, an anti-cancer drug of causing kinase inhibition. Pterostilbene together with sunitinib reduced the cell proliferation marker Ki-67 and increased the expression of caspase-3 when compared to the treatment with sunitinib alone [127]. Furthermore, the inhibition of cyclooxygenase-2 (COX-2) by pterostilbene in non-small cell lung cancer (NSCLC) was shown to stimulate apoptosis and inhibit cell proliferation [128]. Pterostilbene was reported to exert pro-apoptotic and anti-inflammatory properties in T315I-mutated BCR/ABL-positive leukemic cells in an in vitro study. Pterostilbene reduced the protein expression of activated NF-κB and protein kinase B (AKT) which are responsible for the inflammatory response. The pro-apoptotic property of pterostilbene in leukemic cells was reported as pterostilbene increased the protein expression of cleaved caspase-9 and -3 [129]. The anti-inflammatory effects induced by pterostilbene through the modulation of the NF-κB-signalling pathway were also reported in a chemically induced inflammation mouse model. Inflammation induced by 12-O-tetradecanoylphorbol-13-acetate (TPA) treatment stimulates the activation of NF-κB and increases its downstream effectors’ protein expression including that of COX-2 and inducible nitric oxide synthase (iNOS). However, pterostilbene reversed the effect of TPA by suppressing the activation of NF-κB and downregulating the protein expression of COX-2 and iNOS. The same study also showed that topical pterostilbene treatment demonstrated a chemopreventive effect against the development of skin cancer by reducing the number of tumours in a 7,12- dimethylbenz[a]anthracene (DMBA)/TPA-induced skin carcinogenesis mouse model [130].

## 5. Future Prospects of Using Pterostilbene as a Chemopreventive Agent

Chemoprevention research is a rapidly developing medical field that has become increasingly popular. Chemoprevention is one of the promising interventions in fighting cancer. It has the potential to work synergically with other interventions such as cancer screening or chemotherapy to reduce the incidence and mortality rate [131]. There are many obstacles in chemoprevention studies including the cost as these studies need large amounts of funding. A lot of money is needed to search for an effective chemopreventive agent as studies are needed to be conducted for a long duration of preclinical and clinical phases, many subjects need to be involved, the agent’s efficacy and toxic effects on different locations need to be validated, and demographic backgrounds need to be identified [132]. Numerous natural active compounds from plant sources are potential candidates as chemopreventive agents that have been studied extensively [133]. The potential of plant-based bioactive compounds as potential candidates to be developed as effective chemopreventive agents is also supported by numerous epidemiologic studies, in which vegetable and fruit consumption was associated with a lower risk of developing various types of cancer including breast, liver, colorectal and lung cancer [134,135,136]. Even though many plants are edible by humans, the effect of plant bioactive compounds on allergy or toxicological evaluations still cannot be overlooked. Safety assessments are needed, including assessments of exposure levels or the right dose with minimal adverse effects, in short- and long-term applications of plant-derived chemopreventive agents [137].

Hence, pterostilbene as a natural compound with low toxicity can be an ideal chemopreventive agent against cancer development. In addition to apoptosis-inducing properties, pterostilbene has been proven to modulate inflammation via the NF-κB-signalling pathway. The NF-κB-signalling pathway can be targeted in cancer chemoprevention strategies as it plays a crucial role in the development and progression of cancer. This pathway regulates cell death and proliferation and immune and inflammatory responses [138,139]. Pterostilbene has been shown to reduce tumour incidence and multiplicity in azoxymethane-induced colon carcinogenesis in rat models. Further investigations revealed that pterostilbene exerts anti-inflammatory effects to inhibit colon carcinogenesis. A diet containing pterostilbene downregulated the protein expression of NF-κB and its pro-inflammatory mediators such as cyclooxygenase-2 (COX-2) and inducible nitric oxide synthase (iNOS) [140]). Other than the chemopreventive and anti-cancer effects against various types of cancer, pterostilbene has also been reported to exhibit numerous health benefits including anti-bacterial, anti-diabetic, anti-hyperlipidemic, and cardioprotective effects [141,142,143]. A natural analogue of pterostilbene, resveratrol was reported to exert its maximum effect to reduce the levels of the cancer biomarker of plasma insulin-like growth factor-1 (IGF-1) at the optimal dose of 1 g/day administered orally to healthy volunteers. The higher dose of 5 g/day of resveratrol showed a less prominent effect on the reduction in plasma IGF-1 levels [144]. Pterostilbene can be found in numerous crops included in the human diet and consuming pterostilbene-containing food may reduce the risk of developing cancer. However, the consumption of food containing pterostilbene may provide an inadequate amount or an incorrect dose for cancer chemoprevention effects or other health benefits. Hence, a formulation of the pterostilbene compound as a diet supplement and more research to detect and quantify the presence of pterostilbene in the human diet are needed [145]. This may provide information on the right dose or amount of pterostilbene-containing food to be consumed in order to obtain its optimal medicinal benefits, especially in cancer prevention strategies. The introduction of processed food that is enriched with pterostilbene can be an alternative to increasing pterostilbene consumption [141,142,143].

## 6. Conclusions

We conclude that pterostilbene is a potential candidate to be developed as a chemopreventive agent due to its favourable pharmacokinetic properties and various pharmacological activities, especially its ability to induce apoptosis. Further investigation on the modulation of apoptosis by pterostilbene in preventing cancer development at the molecular level should be conducted, although in this review, we highlighted many previous studies with promising results of pterostilbene as an anti-cancer and chemopreventive agent. The findings highlighted are mostly from preclinical studies using cell lines and animal models. Hence, clinical studies of using pterostilbene as an anti-cancer and chemopreventive agent in humans are crucial and relevant. Further research on pterostilbene formulation, drug interactions, and chronic toxicity could lead to the optimal usage of pterostilbene for human health, especially in cancer prevention and treatment.

## Figures and Tables

**Figure 1 ijms-24-09707-f001:**
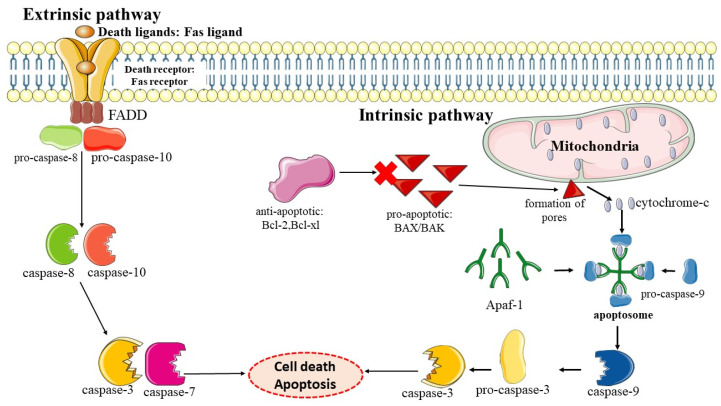
Summary of the extrinsic and intrinsic pathways of apoptosis. The extrinsic apoptosis pathway starts outside of a cell when the death ligands bind to a death receptor at the transmembrane. The intrinsic apoptosis pathway occurs via the mitochondria and can be triggered when stress stimuli are present inside the cells including DNA damage, oncogene activation, and hypoxia. Created using smart.servier.com.

**Figure 2 ijms-24-09707-f002:**
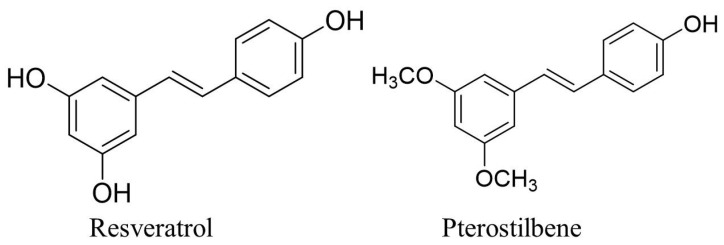
Chemical structures of resveratrol and pterostilbene. Created with ChemDraw Professional 15.0.

**Figure 3 ijms-24-09707-f003:**
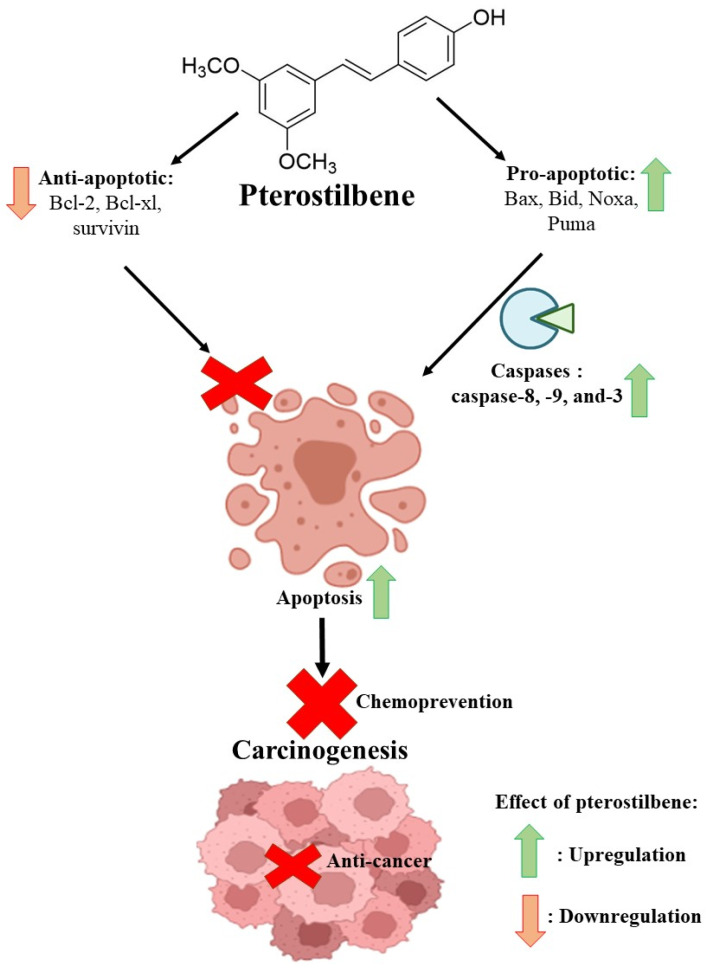
Summary of pterostilbene as a chemopreventive and anti-cancer agent via the modulation of the apoptosis pathway. Created with Biorender.com.

**Table 1 ijms-24-09707-t001:** A summary of pterostilbene as a chemopreventive agent against different types of cancer via the modulation of the apoptosis pathway in animal models.

Type of Cancer	Dose or Concentration and Route of Administration	Outcomes	Effects on Apoptosis
Azoxymethane (AOM) induced aberrant crypt foci (ACF) and colorectal cancer	Concentration: 50 and 200 ppm (dietary)	A diet containing pterostilbene significantly reduced the number of ACF in the colon after 6 weeks of AOM exposure. A diet containing pterostilbene significantly reduced the number of adenomas in an AOM-induced colon carcinogenesis mouse model (23 weeks).	Upregulation of pro-apoptotic proteins Fas, Fas L, Bax, and Bid. Upregulation of apoptotic caspases: cleaved caspase-9, -8, and -3 [112]
Urethane-induced lung adenoma mouse model	Dose: 50 and 250 mg/kg (intraperitoneal injection)	Pterostilbene significantly reduced the tumour multiplicity, volume, and burden.	Upregulation of cleaved caspase-3 [113]
Phosphatase and tensin homolog (Pten) loss in transgenic mouse model of prostate cancer	Concentration: 10 mg/kg diet (dietary)	Mice with a diet containing pterostilbene showed smaller-sized prostate glands and a reduction in the formation of premalignant lesions of prostatic intraepithelial neoplasia (PIN).	Upregulation of pro-apoptotic p27 and cleaved caspase-3 [114]
Diethylnitrosamine (DEN)- and carbon tetrachloride (CCl_4_)-induced hepatocellular carcinoma mouse model	Dose: 100 and 200 mg/kg (intraperitoneal injection)	Pterostilbene inhibited tumour growth as the number and the maximum size of tumours was significantly reduced. Pterostilbene also protects the liver from injury by reducing liver enzymes (AST, ALT, LDH, and ALP)	Increase in the percentage of apoptosis cells [115]
NTCU-induced lung SCC mouse model.	Dose: 10 and 50 mg/kg (intraperitoneal injection)	Pterostilbene reduced cell proliferation and induced cell cycle arrest to inhibit the carcinogenesis of lung cancer.	Upregulation of cleaved caspase-3 [117]

Abbreviations—ACF: aberrant crypt foci; AOM: azoxymethane; AST: aspartate aminotransferase; ALT: alanine transaminase; ALP: and alkaline phosphatase; LDH: lactate dehydrogenase; NTCU: N-nitroso-tris-chloroethylurea.

## Data Availability

Not applicable.

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
