# Peer review of "Potential Chemopreventive Role of Pterostilbene in Its Modulation of the Apoptosis Pathway"

_ijms, 2023, doi:10.3390/ijms24119707_

Round 1

Reviewer 1 Report

After reading the manuscript my major concerns are as follows:

11. It is not mentioned what type of review is presented in this manuscript (narrative, systematic, scoping review?)

22. The paper is missing several outstanding publications (see the list of 10 most important references that should be mentioned in the paper, which describe molecular mechanisms of action of pterostilbene in relation to its anti-proliferative and anti-cancer effects). Please, mention the most important molecular pathways in which pterostilbene is involved.

1: González-Sarrías A, Espín-Aguilar JC, Romero-Reyes S, Puigcerver J, Alajarín M, Berná J, Selma MV, Espín JC. Main Determinants Affecting the Antiproliferative Activity of Stilbenes and Their Gut Microbiota Metabolites in Colon Cancer Cells: A Structure-Activity Relationship Study. Int J Mol Sci. 2022 Dec 1;23(23):15102. doi: 10.3390/ijms232315102. PMID: 36499424; PMCID:PMC9739882.

2: Jin J, Shan Y, Zhang L, Wu Z, Wu S, Sun M, Bao W. Pterostilbene Ameliorates Fumonisin B1-Induced Cytotoxic Effect by Interfering in the Activation of JAK/STAT Pathway. Antioxidants (Basel). 2022 Nov 28;11(12):2360. doi: 10.3390/antiox11122360. PMID: 36552567; PMCID: PMC9774891.

3: Wawszczyk J, Jesse K, Kapral M. Pterostilbene-Mediated Inhibition of Cell Proliferation and Cell Death Induction in Amelanotic and Melanotic Melanoma. Int J Mol Sci. 2023 Jan 6;24(2):1115. doi: 10.3390/ijms24021115. PMID: 36674631; PMCID: PMC9866175.

4: Monceaux K, Gressette M, Karoui A, Pires Da Silva J, Piquereau J, Ventura-Clapier R, Garnier A, Mericskay M, Lemaire C. Ferulic Acid, Pterostilbene, and Tyrosol Protect the Heart from ER-Stress-Induced Injury by Activating SIRT1-Dependent Deacetylation of eIF2α. Int J Mol Sci. 2022 Jun 14;23(12):6628. doi: 10.3390/ijms23126628. PMID: 35743074; PMCID: PMC9224298.

5: Hojo Y, Kishi S, Mori S, Fujiwara-Tani R, Sasaki T, Fujii K, Nishiguchi Y, Nakashima C, Luo Y, Shinohara H, Kuniyasu H. Sunitinib and Pterostilbene Combination Treatment Exerts Antitumor Effects in Gastric Cancer via Suppression of PDZD8. Int J Mol Sci. 2022 Apr 4;23(7):4002. doi: 10.3390/ijms23074002. PMID:35409367; PMCID: PMC8999764.

6: Yi M, Wang G, Niu J, Peng M, Liu Y. Pterostilbene attenuates the proliferation and differentiation of TNF-α-treated human periodontal ligament stem cells. Exp Ther Med. 2022 Apr;23(4):304. doi: 10.3892/etm.2022.11233. Epub2022 Feb 22. PMID: 35340874; PMCID: PMC8931590.

7: Wang Z, Wang T, Chen X, Cheng J, Wang L. Pterostilbene regulates cell proliferation and apoptosis in non-small-cell lung cancer via targeting COX-2. Biotechnol Appl Biochem. 2023 Feb;70(1):106-119. doi: 10.1002/bab.2332. Epub2022 Mar 11. PMID: 35231150.

8: Wawszczyk J, Jesse K, Smolik S, Kapral M. Mechanism of Pterostilbene-Induced Cell Death in HT-29 Colon Cancer Cells. Molecules. 2022 Jan 7;27(2):369. doi: 10.3390/molecules27020369. PMID: 35056682; PMCID: PMC8779997.

9: Kawakami S, Tsuma-Kaneko M, Sawanobori M, Uno T, Nakamura Y, Matsuzawa H, Suzuki R, Onizuka M, Yahata T, Naka K, Ando K, Kawada H. Pterostilbene downregulates BCR/ABL and induces apoptosis of T315I-mutated BCR/ABL-positive leukemic cells. Sci Rep. 2022 Jan 13;12(1):704. doi: 10.1038/s41598-021-04654-1. PMID: 35027628; PMCID: PMC8758722.

10: Chen RJ, Lyu YJ, Chen YY, Lee YC, Pan MH, Ho YS, Wang YJ. Chloroquine Potentiates the Anticancer Effect of Pterostilbene on Pancreatic Cancer by Inhibiting Autophagy and Downregulating the RAGE/STAT3 Pathway. Molecules. 2021 Nov 8;26(21):6741. doi: 10.3390/molecules26216741. PMID: 34771150; PMCID:PMC8588513.

Author Response

Dear Dr/Prof,

Thank you very much for your comments and suggestions on our manuscript. We have amended our manuscript based on your comments and suggestions. We have attached a file of responses to the comments and suggestions (as below). 

Reviewer 2 Report

·    The focus of the review is chemoprevention and the activity of the compound pterostilbene in modulating apoptosis.  The review describes in detail, apoptosis with the various mechanisms.  Authors mention only 5 works for chemoprevention and only two molecules, curcumin and honokiol, are described. Other natural compounds are not used for chemoprevention?

·     It is described the role of  pterostilbene in chemoprevention,  but very little about its involvement in the regulation of apoptotic mechanisms. This part should be described better.

·         The effects of pterostilbene on the regulation of non-apoptotic cell deaths are also known. I believe that for a complete description of the compound this part should be added.

Author Response

Dear Dr/Prof,

Thank you very much for your comments and suggestions on our manuscript. We have amended our manuscript based on your comments and suggestions. We have attached a file of responses to the comments and suggestions (as below). 

Best regards

Reviewer 3 Report

Dear colleagues!

Respected team of authors in this research describe the advantages in the possible use in the fight against cancer of a natural bioactive substance from the group of stilbenoids - pterostilbene.

Pterostilbene is a natural chemo protective compound with various pharmacological properties such as antioxidant, anti-proliferation, and anti-inflammation. Moreover, the potential chemo protective effect of ptrostilbene to induce apoptosis in eliminating the mutated cells or preventing the progression of premalignant cells to cancerous cells should be explored as a chemo preventive agent. Hence, in the review, authors discussed the role of pterostilbene as a chemoprotective agent against various types of cancer via its modulation of the apoptosis pathway at the molecular levels.

The article is of interest in many respects and certainly worthy of publication in such a respectable journal.

At the same time, I would like to point out to the authors some additions that need to be made.

1. Authors write: Apart from that, pterostilbene can be found in natural sources, including grapes (Vitis vinifera), berries, and peanuts.

The largest amount of the studied biologically active compound was found in the grapes, and especially in the buds of the Vitis amurensis, this must be mentioned in the article. Also, the second most important source of ptrostilbenes is Radix polygoni multiflori.

2. Authors write: The biosynthesis of pterostilbene from the resveratrol can occur both in vitro and in planta of grapevine (Vitis vinifera) leaves that catalyse by the resveratrol O-methyltransferase under the stress stimulus such as fungal infection (Plasmodia viticola).

The authors need to point out that very interesting studies have been carried out with grape buds (Vitis amurensis) as well as with its inflorescences, which show many times more pterostilbenes than other plant matrices.

Best regards

Author Response

(The authors gave the same response as above.)

Reviewer 4 Report

In this manuscript, authors reported a review for the potential chemopreventive role of pterostilbene in its modulation of the apoptosis pathway. Although the author structured and wrote the manuscript well, this manuscript cannot be submitted as an article because there is no direct research content of the author. This manuscript also has several problems and lacks as following:

1.      Since this manuscript is not a submission of the author’s original research contents, it should be changed to a review instead of an article.

2.      This manuscript devotes too many references to general knowledge of chemoprevention and apoptosis other than pterostilbene.

3.      Authors should add the contents related to apoptotic molecular mechanism of pterostilbene.

4.      To help reader’s understanding, please draw a schematic diagram of pterostilbene’s chemoprevention.

5.      Please describe how much and how we should have pterostilbene to get the significant chemopreventive in human?

6.      The content of the conclusion is too simplistic and lacks a summary that represents the whole content. 

Author Response

(The authors gave the same response as above.)

Round 2

Reviewer 1 Report

No further concerns to the paper.

Reviewer 2 Report

The authors have added some required information, improving the manuscript

Reviewer 4 Report

I read carefully this revised manuscript entitled “Potential chemopreventive role of pterostilbene in its modulation of the apoptosis pathway”. Authors made responses and corrections for the comments I had pointed out. However, this manuscript still has several problems as following:

1.      In the keywords, please remove ‘natural product’.

2.      In the line 158 of page 4, authors should correct ‘Bcl-2 homologues antagonist (Bak)’ to Bcl-2 homologues antagonist/killer (Bak1).

3.      In order to avoid confusion, it is desirable to move the word ‘Intrinsic pathway’ inside the cell membrane in the Figure 1.

4.      In the lines 243-246, authors should revise the sentence to ‘In addition, many other natural compounds including epigallocatechin gallate (EGCG), genistein, luteolin, lycopene, ellagic acid and lupeol have been extensively studied for their chemoprevention and anticancer effects against various types of cancer.’

5.      In the lines 260-263, authors should revise the sentence to ‘For example, the various stilbene compounds including polydatin, resveratrol, 2,3,5,4’-tetrahydroxystilbene-2-O-β-D-glucoside and pterostilbene have been detected in a traditional Chinese medicine, Radix Polygoni multiflori [90,91]’. And please don't use italic for the 'Radix Polygoni multiflori'.  

6.      In the line 265, please remove ‘also’.

7.      In the line 267, authors should revise ‘a herb’ to ‘a herbal medicine’ or ‘a folk medicine’.

8.      For the sentence of lines 274-275, authors should correct that to ‘Both pterostilbene and resveratrol are the well-known and widely studied stilbenes that share almost similar chemical structure with other stilbenes, consisting of a C6-C2-C6’.

9.      Authors must check the English grammar of the entire manuscript.